# Home Respiratory Care: Design of a Prototype for Continuous Measurement at the Nasal Septum

**DOI:** 10.3390/healthcare10020318

**Published:** 2022-02-08

**Authors:** Roshini Narayanan, Evan Bender, Raphael Chernoff, Luis Mendoza, Samuel Bernstein, Emma Turner, Stephen Mathai, Constanza Miranda

**Affiliations:** 1Department of Biomedical Engineering, Johns Hopkins, Baltimore, MD 21218, USA; ebender6@jhu.edu (E.B.); rcherno2@jhu.edu (R.C.); lmendoz2@jhu.edu (L.M.); sberns12@jhu.edu (S.B.); eturne35@jhu.edu (E.T.); 2School of Medicine, Johns Hopkins, Baltimore, MD 21218, USA; smathai4@jhmi.edu

**Keywords:** aging, chronic pulmonary disease, oxygen therapy, supplemental oxygen, self-management, home care, quality of life, sensors, pulse oximetry

## Abstract

Chronic respiratory diseases have been on the rise, especially due to COVID-19, extreme air pollution, and other external circumstances. Millions of people around the world suffer from progressive lung diseases and require supplemental oxygen therapy to maintain blood oxygen (SpO2) levels above 90% to prevent hypoxic episodes that can lead to further organ damage. Today, these chronic episodes are more prevalent in aging populations suffering from Chronic Obstructive Pulmonary Disorder (COPD). Existing SpO2 measurement equipment, designed to assist with treating COPD at home, are suboptimal as they cannot measure SpO2 levels continuously, meaning supplemental oxygen devices are unable to adjust oxygen flow rates to the patient’s needs. These discrepancies can result in hypoxic episodes of blood oxygen levels below 90%. Following this need, our team demonstrates preliminary results of the novel placement of a SpO2 sensor in the nasal septum to allow for comfortable and sustained SpO2 measurement. This will improve the experience of home-respiratory care with continuously obtained data from a novel location.

## 1. Introduction

Long term oxygen therapy (LTOT) is prescribed as care for over 1.5 million progressive lung disease patients in the United States [1]. COPD is the most common of these diseases, caused by the inflammation of airways and destruction of alveoli (air sacs) in the lungs. COPD has an incidence of 5% and is the 4th leading cause of death in the United States [2,3]. It causes chronic hypoxic states, which occur when SpO2 levels fall below 90% to result in low blood oxygen circulation. Furthermore, sustained hypoxia can result in worsening of respiratory symptoms such as coughing, difficulty breathing, and more dangerously: the development of additional health conditions, including pulmonary hypertension, secondary polycythemia, systemic inflammation, and skeletal muscle dysfunction [4]. Thus, COPD patients using supplemental oxygen must consistently maintain their SpO2 level above 90% to prevent instances of hypoxia in order to reduce the severity of COPD symptoms and complications from sustained hypoxic episodes. For patients, these episodes arise during daily activities, most likely to occur during sleep and physical exertion, and they must be addressed immediately through the increase in oxygen flow rate to preserve health [4]. 

LTOT is essential for maintaining SpO2 at a safe level for patients, by increasing overall oxygen availability. Patients receive oxygen via a device such as a refillable tank or portable concentrator, which both have adjustable flow rates that are set in accordance with the patient’s prescribed flow rate and current need [1]. These flow rates are not automatically adjustable, and they only provide a single, static rate of oxygen that may not always be the required amount to keep SpO2 level above 90% [1]. To understand if oxygen flow rates are inadequate, commercially available SpO2 sensors are utilized to benchmark the patient’s blood oxygen level as a percentage. SpO2 monitors are bulky and generally inadequate for continuous monitoring and optimized patient care [5]. They are obstructive and limit patients’ access to vital data. Sensor location is the main limitation of these devices. The most common locations for SpO2 sensors are at the fingers and wrist. However, both locations are subject to considerable movement during everyday activities. This introduces considerable noise and error into pulse oximetry readings, resulting in largely inaccurate data pulled from the devices [5]. Finger-based sensors are additionally ineffective because they can limit everyday activities due to their obstruction of the patient’s grip. They are bulky and fall off easily, meaning that continuous monitoring is not possible with current pulse oximeter technology [5]. 

The nasal septum is an attractive location for SpO2 measurement because it maintains many of the requirements for accurate pulse oximetry. The nasal septum contains a blood-rich region supplied by the facial arteries that perfuse blood to the area through the septal arteries [6]. This provides a suitable location to measure blood oxygen levels with pulse oximetry. More importantly, the nose is considerably less susceptible to motion impact in contrast to locations such as the hand or wrist, making it a viable location to reduce noise and external signal disruption [7]. This is an important consideration because, as mentioned previously, motion can introduce considerable error into device accuracy [5]. In addition, the nasal location will not interfere with everyday activities as common finger based or ear-based sensors do, since the sensor can be easily clipped and maintained at the septum, which is not usually disturbed in daily life [7].

Ultimately, it becomes clear that current pulse oximeters are unsuitable for continuous measurement, and lack of continuous data can result in ignorance to daily hypoxic episodes and worsening conditions. Hypoxic episodes can occur without warning, and symptoms can be difficult for patients to identify or invisible, so self-adjustment to drops in SpO2 are nearly impossible [8]. It is incredibly important to catch hypoxic episodes early so that they are not sustained to the point of further organ damage, meaning that continuous SpO2 measurement is an essential need that can be fulfilled with nasal septum placement. The objective of this study is to obtain preliminary results indicating the feasibility of SpO2 measurement at the nasal septum, utilizing a prototype manufactured and tested by the team. 

## 2. Materials and Methods

Our innovation involves the placement of a SpO2 sensor at the nasal septum. The sensor design is a transmittance based SpO2 sensor, meaning light passes through the nasal septum and is received on the other side [9]. The following metrics were used as benchmarks for a successful implementation of the sensor at the septum. These requirements are tested for fulfillment in the Results section. The device must:Accurately detect heartbeat via two wavelengths of light passing through the nasal septumHave a stable placement to reduce the impact of motion on measurements. For this to be accomplished, the sensor must not move while it is being worn.Be comfortable for extended use, ensuring patients can wear the device throughout their entire day with minimal discomfort and impediment to their daily activities.

Prototype Development:

Figure 1a and Figure 2 show the physical representation of the prototype: a SpO2 sensor placed in the nasal septum. 

This prototype used transmittance oximetry to capture heartbeats at the nasal septum, as this method is shown to be more accurate and reliable than reflectance based oximetry [10,11,12]. Transmittance oximetry relies on alternating pulses of light, at different wavelengths, to pass through a part of the body and be received on the other side. SpO2 measurements are determined by utilizing the difference in light absorption between oxygenated and deoxygenated hemoglobin [9]. Oxygenated hemoglobin absorbs more infrared light (940 nm) and less red light (660 nm) than hemoglobin alone [9]. These pulses are supplied by the heartbeat, indicating the important correlation between heartbeat and SpO2 data collection.

The team developed a basic design for a transmittance-based sensor to be placed at the nose. The design has two prongs, located at the end of a nose clip, to be placed in the nose. One prong contains red and near infrared (NIR) LEDs, with wavelengths of 660 and 940 nm, respectively, which are in the ideal range for SpO2 measurement [9]. The opposing prong contains a photodiode, which receives the light emitted from both LEDs. The circuitry is controlled by an Arduino UNO microcontroller. The microcontroller pulses the LEDs at a rate of 8 Hz, and it receives the signal from the photodiode.

The two prongs are made from sprung steel, which can easily be bent to fit for patients while providing sufficient pressure for recording. The sensor was coated in EcoFlex, a soft, skin-safe polymer, for patient comfort and durability. The final design is depicted in Figure 1c. The physical prototype is shown in Figure 2, properly placed on the wearer.

To test the feasibility of the nasal septum as an area to obtain SpO2 measurements, a testing protocol was used to determine the possibility of successful implementation. A team member positioned the prototype (Figure 1c) containing functioning SpO2 measurement technology at their nasal septum, connected to a microcontroller that would process the data to the computer. Code was written to obtain the data in an easily legible format and create a graph using this data. 

## 3. Results

Utilizing the sensor worn at the nose, data from NIR and red spectrums was collected via the microcontroller. The successful collection of desired data, shown in Figure 3, validates the usage of the septum as a viable collection site for SpO2. The cyclical nature of the absorbance patterns corresponds to the user’s heart rate; the waves correspond to the heart beats, and the troughs represent the flow of deoxygenated blood, while the peaks correspond to oxygenated blood pulses. The x-axis of Figure 3 is the length of time passed, and the y-axis is the level of light transmitted by the NIR and Red LEDS. Transmitted light corresponds directly with the oxygenation level of hemoglobin as absorbance changes based on the presence of oxygen. The ratio between the absorbance of oxygenated and deoxygenated hemoglobin is then calculated by the connected microprocessor, which utilizes the Beer–Lambert Law to calculate SpO2 level percentage [9]. The Beer–Lambert Law is shown as A = εbc, where A refers to absorbance, ε refers to molar absorptivity, b is the length of the light path emitting from the LED, and c is the concentration [9]. Using the data taken from the wavelengths of the LED and IR light units in the SpO2 sensor, the microprocessor is able to convert the NIR and Red spectrums into a viewable blood oxygen percentage value. 

## 4. Discussion

To mediate gaps in supplemental oxygen treatment due to lack of personalized care, the team proposes a novel SpO2 sensor placement in the nasal septum to continuously monitor blood oxygen levels in patients. The team was able to validate and verify the design of the prototype with tests that confirmed usable SpO2 data could be obtained from the nasal septum. Shown in Figure 3, our prototype can distinguish pulsatile wave data for both the NIR and red spectrum, meaning that viable blood oxygen data can be obtained through the conversion using Beer–Lambert’s Law. With the data received from this system, the prototype can be modified into a clip that is continuously and comfortably placed at the nasal septum of elderly populations with respiratory illnesses. As mentioned previously, hypoxic episodes may not always manifest themselves in outward physical symptoms such as coughing, but they still cause long-term damage to organs if occurring often in the patient’s daily life. Continuous SpO2 level measurement will indicate when these hypoxic states occur across long periods of time and can inform the patient and their physician that there is still risk for chronic pulmonary disease progression. Continuous monitoring will also serve useful for clinicians looking to study elderly patients’ lifestyle and recommend the avoidance of certain activities that may cause hypoxic episodes unbeknownst to the patient themselves. This region does not fall victim to daily disturbances, so the sensor will monitor SpO2 levels constantly and can, in the future, be integrated with current supplemental oxygen equipment to modulate oxygen flow rate based on blood oxygen levels, which may fluctuate with daily activities such as walking up the stairs. Overall, the results of this experiment showed that there is viability for SpO2 sensors to be placed at the nasal septum to obtain continuous and accurate readings of blood oxygen levels. The sensor’s materials and design contributed to the comfortability of its placement and provided a sturdy packaging for the electronics to be enclosed in. 

These technical results did not take comfort testing into account, ultimately indicating that further modifications may be done to optimize the exterior packaging for the least burden to the patient and ensure that motion will not impact SpO2 data acquisition. For future directions, the nose-based sensing location allows for easy integration into existing equipment such as nasal cannulas. The novel placement of the sensor can be innovatively utilized to modulate the level of oxygen flow in an informed fashion, delivering modulated amounts of oxygen to maximize their level of care and ensure a healthy blood oxygen circulation. Ultimately, this design offers numerous future directions to better serve the aging population at risk for acquiring and sustaining progressive lung diseases. 

## Figures and Tables

**Figure 1 healthcare-10-00318-f001:**
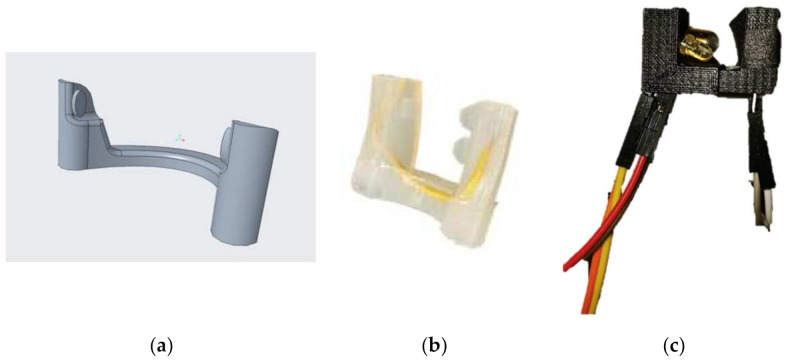
(**a**). This figure shows the nasal SpO2 sensor CAD rendering. The upturned edges indicate where the LEDs and IR light emitters will be placed, and the curved center will follow the curve of the nasal septum. (**b**) shows the physical concept with the protective Ecoflex polymer binding juxtaposed against the working sensor prototype in (**c**). This (**c**) prototype is optimized for comfort and durability, and it achieved the SpO2 readings mentioned in the Results section.

**Figure 2 healthcare-10-00318-f002:**
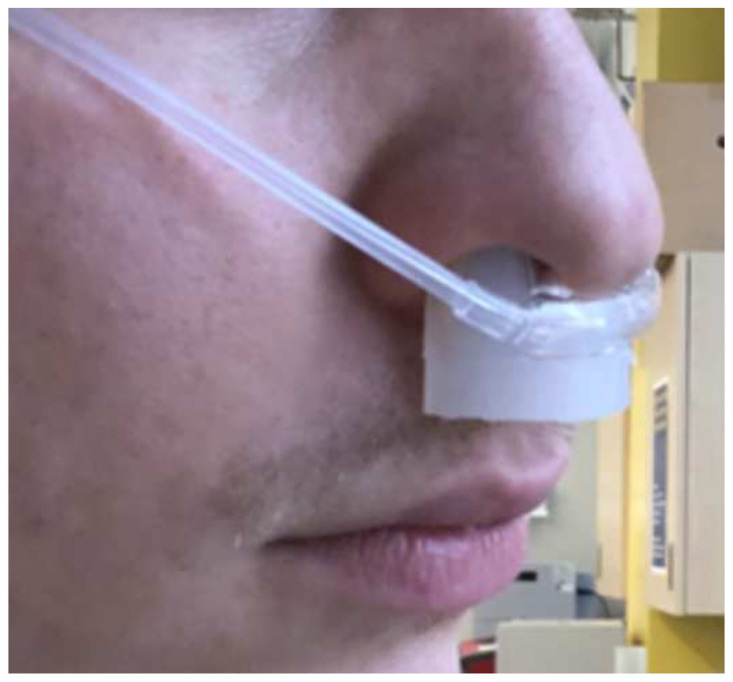
This figure shows the placement of the nasal septum sensor in an imaged nose, indicating the positioning of the device within the nasal cannula. This placement was modeled on a team member.

**Figure 3 healthcare-10-00318-f003:**
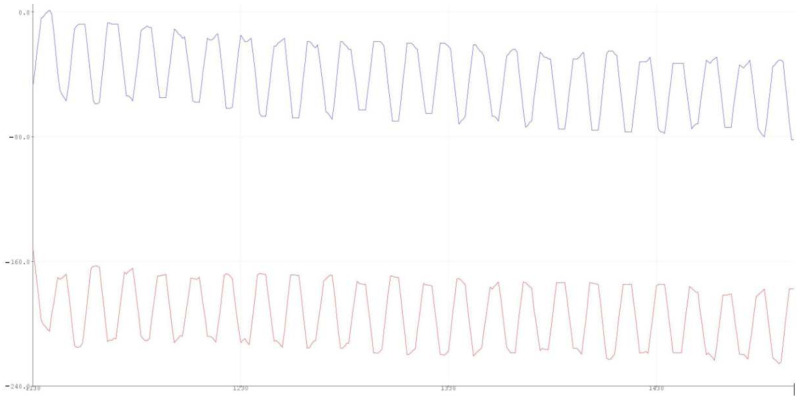
This figure shows the results of the SpO2 sensor testing at the nasal septum. The graph shown in blue refers to the NIR spectrum, and the graph shown in red refers to the red spectrum.

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
