# Peer review of "Home Respiratory Care: Design of a Prototype for Continuous Measurement at the Nasal Septum"

_healthcare, 2022, doi:10.3390/healthcare10020318_

Round 1

Reviewer 1 Report

In the manuscript “Home respiratory care: enhancing oxygen therpy using continuous measurement at the nasal septum”, R. Narayanan et al. presents a novel placement of an SpO2 sensor (in the nasal septum) promising a more comfortable and sustained SpO2 measurement.

The idea is of interest in the field. Nevertheless, some important concerns are in the mind of this reviewer:

1) The manuscript needs to be revised by an English native speaker. It results often difficult to understand the messages authors seek to communicate. Here below some examples:

Line 26: “of these diseases in COPD, a disease” >> “COPD represents the most common cause of TLTO, and is caused by… “

Line 28: “distrucion of alveoli” >> “disruption of alveoli”

Line 28-29: “impacting roughly 5% of the US population and is the 4th leading cause of death in the United States” >> “It has an incidence of 5% and is the 4th cause of death in the United States”

Line 42 “by increasing the oxygen available to the lungs with each breath” >> “by increasing oxygen availability.”

Line 43-45: “flow rates…flow rate…flow rates” repetition

Line 50: “sensor location is the one of the primary causes for this…” “sensors location is the main limit…”

Moreover, there is an appropriate use of abbreviations (Line 106: IR)

2) The device is interesting, but I do not understand how the O2 measurement has been validated.

3) Material and Methods are crucial. This part needs to be implemented.

Author Response

Response to Reviewer 1:

  1. It results often difficult to understand the messages authors seek to communicate. Here below some examples:

 We apologize for the grammatical issues with our submission and have carefully revised it based on the reviewer’s recommendations. The entire team is composed of native English speakers and we have reviewed and edited the submission to be more understandable. Abbreviations have been explained and corrected throughout the submission. 

  1. The device is interesting, but I do not understand how the O2 measurement has been validated.

The oxygen measurement has been validated through the Results section, specifically Figure 5. First, the figure shows how cyclical plots are obtained, which correspond to the heartbeat. The pulsatile heart rate is essential to SpO2 measurement because these sensors rely on the increase of red light absorbance with each heartbeat to obtain oxygen readings with each new cycle of blood to the sensory area. Therefore, the troughs and peaks indicate that the sensor is correctly picking up on pulses of blood flow. Second, the blue plot refers to the NIR spectrum and the red plot refers to the red spectrum. Each of these references how the SpO2 sensor is able to obtain readings, and the wave nature of the plot again shows how blood pulses influence the light absorbance, which directly corresponds to blood oxygen level. A more detailed explanation is given in the Results section, as well as an overview in the Materials and Methods section: “A pulse oximeter measures the SpO2 by utilizing the difference in light absorption between oxygenated and deoxygenated hemoglobin [9]. Oxygenated hemoglobin absorbs more infrared light (940 nm) and less red light (660 nm) than hemoglobin alone.” Because the reviewer noted lack of clarity in the validation of oxygen measurement, our Results section has been reviewed and elaborated for improved understanding. 

  1. Material and Methods are crucial. This part needs to be implemented.

Our Materials and Methods section reflects the prototype development and how this prototype was implemented to gain viable oxygen measurements from the nasal septum. We have modified our Methods section to reflect the implementation of the device, but it is slightly unclear if the reviewer is referring to something larger. We would love to get more information about this point so that we can make the necessary revisions!

Reviewer 2 Report

The manuscript dealt with an interesting issue. However, there needs to be more improvement in the manuscript. 

  1. The study’s objective was not clearly mentioned: development of prototype or feasibility of implementation of the prototype in the nasal septum?
  2. If the feasibility was the object, figure1a & 2 need to be shown in the method section.
  3. The authors should describe the method and result section in more detail. The results should be described as well as presented with figures.   
  4. Figure5 was too blurred to be read.
  5. As the discussion is too short, the manuscript needs to be resubmitted as a letter or brief communication rather than a research article. 

Author Response

Reviewer 2:

  1. If the feasibility was the object, figure1a & 2 need to be shown in the method section.

Feasibility of the prototype as a nasal septum-based SpO2 sensor was the objective of our study. We have moved Figures 1a and 2 to the methods section as the reviewer suggested. 

  1. The authors should describe the method and result section in more detail. The results should be described as well as presented with figures.   

We have increased the level of detail in our methods and results section as the reviewer suggested. Figure 5 has been explained in more detail to reflect the results of our study, showing how the feasibility of nasal septum SpO2 placement is fulfilled. 

  1. Figure5 was too blurred to be read.

We apologize that Figure 5 was too blurry. Unfortunately as our team is currently on holiday break, we are not able to obtain higher resolution images of the data and will require three weeks’ time to upload a non-blurry image. A supplemental image with more description has been added as a placeholder until this image can be replaced. 

  1. As the discussion is too short, the manuscript needs to be resubmitted as a letter or brief communication rather than a research article. 

Because this is the first paper this team has submitted to, we believed that ‘Article’ was the correct submission type. We agree with the reviewer that ours is too short and therefore, we have decided to resubmit this as a brief communication and have consequently changed the header of our submission to read ‘Brief Communication’. If there are other guidelines that must be followed to resubmit in this fashion, please let us know and we are happy to comply!